# Expanding the Chemical Space of Arsenicin A-C Related Polyarsenicals and Evaluation of Some Analogs as Inhibitors of Glioblastoma Stem Cell Growth

**DOI:** 10.3390/md21030186

**Published:** 2023-03-17

**Authors:** Jacopo Vigna, Denise Sighel, Emanuele Filiberto Rosatti, Andrea Defant, Michael Pancher, Viktoryia Sidarovich, Alessandro Quattrone, Ines Mancini

**Affiliations:** 1Laboratory of Bioorganic Chemistry, Department of Physics, University of Trento, Via Sommarive 14, 38123 Trento, Italy; 2Laboratory of Translational Genomics, Department of Cellular, Computational and Integrative Biology (CIBIO), University of Trento, Via Sommarive 9, 38123 Trento, Italy; 3High Throughput Screening (HTS) and Validation Core Facility, Department of Cellular, Computational and Integrative Biology (CIBIO), University of Trento, Via Sommarive 9, 38123 Trento, Italy

**Keywords:** marine metabolite, polyarsenical, As-S cage, calculated NMR spectrum, arsenic trioxide, antitumor, glioblastoma, cancer stem cell, hypoxia, ADME prediction

## Abstract

The marine polyarsenical metabolite arsenicin A is the landmark of a series of natural and synthetic molecules characterized by an adamantane-like tetraarsenic cage. Arsenicin A and related polyarsenicals have been evaluated for their antitumor effects in vitro and have been proven more potent than the FDA-approved arsenic trioxide. In this context, we have expanded the chemical space of polyarsenicals related to arsenicin A by synthesizing dialkyl and dimethyl thio-analogs, the latter characterized with the support of simulated NMR spectra. In addition, the new natural arsenicin D, the scarcity of which in the *Echinochalina bargibanti* extract had previously limited its full structural characterization, has been identified by synthesis. The dialkyl analogs, which present the adamantane-like arsenicin A cage substituted with either two methyl, ethyl, or propyl chains, were efficiently and selectively produced and evaluated for their activity on glioblastoma stem cells (GSCs), a promising therapeutic target in glioblastoma treatment. These compounds inhibited the growth of nine GSC lines more potently than arsenic trioxide, with GI_50_ values in the submicromolar range, both under normoxic and hypoxic conditions, and presented high selectivity toward non-tumor cell lines. The diethyl and dipropyl analogs, which present favorable physical-chemical and ADME parameters, had the most promising results.

## 1. Introduction

Isolated from the New Caledonian poecilosclerid sponge *Echinochalina bargibanti*, we reported arsenicin A (**1**) as the first polyarsenical compound in nature, characterized by a peculiar adamantane-type structure [1]. More recently, we isolated and identified from the same sponge extract the minor sulfur metabolites arsenicin B (**2**) and arsenicin C (**3**), presenting a nor-adamantane scaffold containing the unusual arsenic–arsenic bond (Figure 1) [2]. The structural elucidation of these peculiar metabolites was absolutely not trivial in the standard natural product study context. Indeed, the diagnostic value of experimental NMR analysis was very poor, because of the lack of reliable references and the presence of very few signals, further limited by symmetry elements, as in the case of arsenicin A [1]. Later, density functional theory (DFT)-NMR [3] and DFT-IR calculations [4] proved to be efficient predictive methods, successfully applied to the study of arsenicin B and C [2] and of future related structures.

Arsenical compounds are among the oldest therapeutic agents used for the treatment of a variety of human diseases [5,6]. Although known for its toxicity and carcinogenicity, arsenic is present in the chemical structure of known drugs. Among them, examples are represented by Salvarsan applied to treat syphilis, arsenic trioxide (ATO), and other forms of As-S inorganic compounds used for a long time in Chinese medicine. ATO, as Trisenox^®^ injectable formulation, was approved by the Food and Drug Administration (FDA) in 2000 for the treatment of acute promyelocytic leukemia (APL) [7]. After its FDA approval, ATO was the object of numerous studies aimed at investigating its mechanism of action and at evaluating its activity in solid tumors, such as gliomas, hepatocellular carcinoma, or lung cancer [7,8]. Unfortunately, despite the promising in vitro results, often ATO concentrations reaching solid tumors are low and systemic toxicity as a side effect has been observed, limiting its transition from bench to bedside for solid tumors. This is particularly true for glioblastoma, where despite the promising in vitro results, ATO success failed to translate into the clinic, as several clinical trials on glioma patients testing ATO alongside standard-of-care therapies failed to yield significant improvements in patients’ overall survival [9]. This lack of clinical activity has been often attributed to the presence of the blood–brain barrier (BBB), which ATO can only sparsely pass, failing to reach effective concentrations inside the tumor.

Despite natural arsenicin A–C having been first reported as potent antibacterial agents [1,2], biological attention has moved on to the evaluation of their antitumor activity. Synthetic arsenicin A and related polyarsenicals (Figure 2) have been evaluated for their antitumor effect both on leukemia and solid tumors in vitro and have been shown to be more potent than ATO. For example, arsenicin A and its derivative monosulfur tetraarsenical **4** (Figure 2) exhibit greater anti-proliferative activity in solid tumor cells compared to ATO [10,11]. Moreover, the tetraarsenical adamantane structure **5**–**7** (Figure 2) obtained by Mancini et al. and selected for an in vitro screening on the NCI- full panel of human cancer cell lines showed significantly higher cytotoxicity compared to ATO against all the various cancer cell lines tested [12], with compound **7** being particularly effective in inhibiting growth of solid tumor cell lines. Based on these promising results, polyarsenical compounds are undoubtedly worthy of further study and characterization of their activity on solid tumors.

Regarding solid tumors, glioblastoma (GBM) is the most common malignant primary brain tumor in adults and among those with the lowest survival expectancy. Despite treatments for this tumor being extremely aggressive, including extensive surgical resection followed by cycles of radiotherapy and chemotherapy, patients survive, on average, 15 months after diagnosis and only 5.5% of them survive more than 5 years [13]. Very limited pharmacological treatment options are currently available for newly diagnosed GBMs and involve the administration of temozolomide (TMZ), an alkylating agent that causes DNA-damage-inducing double-strand breaks. Based on these premises, there is a high and urgent interest in finding novel and potentially effective agents to treat this type of tumor. A major problem in GBM treatment is that the tumor population is highly heterogeneous. Within the different cell types composing the tumor mass, the subgroup of glioblastoma stem cells (GSCs) plays a fundamental role in every malignant feature of the tumor, such as tumorigenesis, progression, invasion, and aggressiveness [14]. Since GSCs have been also shown to be resistant to currently available therapies and to be responsible for disease recurrence, the development of agents specifically targeting them emerges as an important and promising new strategy for GBM therapy that could lead to tumor eradication.

In our search for new effective agents able to target GSCs, we focused our attention on polyarsenicals inspired by the structures of *E. bargibanti* metabolites. Strong in the skills acquired so far in the structural elucidation and in the synthesis of these compounds, we have here expanded the medicinal chemistry space of arsenicin A-C related molecules by synthesizing new analogs, which have been tested for their antitumor activity on GSCs in comparison with ATO. Moreover, we report the identification by synthesis of the very minor metabolite arsenicin D present in the *E. bargibanti* sponge extract.

## 2. Results and Discussion

### 2.1. Chemistry

#### 2.1.1. Synthesis of Alkyl Polyarsenical Analogs

According to the method reported by Keppler and coworkers [15], compounds presenting a dialkyl-substituted tetraarsenic adamantane cage can be synthesized in one step by heating the suitable carboxylic acid and anhydride with ATO and anhydrous potassium carbonate. The synthesis of the dimethyl and diethyl analogs **7** and **8**, whose structure has been confirmed by X-ray diffractometric analysis, has been reported starting from propionic acid/propionic anhydride and butyric acid/butyric anhydride, respectively (Figure 1) [15]. We have already previously used compound **7**, both as a model molecule to support the MS fragmentation of natural arsenicin A [1] and in the in vitro evaluation of tetraarsenicals as antitumor agents [12].

In order to expand this homologous series, the new dipropyl compound **9** was obtained starting from valeric acid and valeric anhydride (Figure 1). Its structure was elucidated, obtaining data in line with those of the extensively studied lower homologs. In detail, a high-resolution APCI-MS experiment carried out in positive ion mode confirmed the molecular composition C_8_H_16_As_4_O_4_. Structural characterization of **9** by NMR analysis was facilitated by the increased contribution of the carbon and hydrogen atoms present in the molecule, as well as of the experience and reference data acquired in the study of similar structures, both natural and synthetic, since the first report on arsenicin A [1]. Additionally, a nice agreement was observed between the experimental and DFT-calculated IR spectra of both compounds **8** and **9** (Appendix A). Furthermore, the low-frequency active transitions assigned to As-O stretching (797 and 796 cm^−1^ for **8** and **9**, resp.) and to As-C stretching (727 and 731 cm^−1^ for **8** and **9**, resp.) agreed with the values previously reported for dimethyl analog **7** [1].

Of note, the reactions with homologous combinations of carboxylic acid and corresponding anhydrides show an increasing selectivity in product formation. This phenomenon can be ascribed to a growing steric effect pushing reaction to produce compound **7** as the very major product and **8** and **9** as single products. Furthermore, the steric hindrance was observed to be decisive in moving from linear to bulkier alkyl reagents (branched, as isobutyric and isovaleric, or aromatic acid/anhydride) which did not allow the formation of the tetraarsenic adamantane structures. Conversely, the use of acetic acid/acetic anhydride for the production of the tetraarsenic cage with no chain provided a series of four products, whose relative ratios were not altered by carrying out the reaction using conventional heating or replacing it by microwave irradiation [12].

#### 2.1.2. Synthesis of Arsenic–Sulfur Compounds **10**–**12**

By reacting arsenicin A with Na_2_S in water, the formation of a series of products presenting sulfur atoms in place of the oxygen atoms in the adamantane cage was reported [11]. In detail, the trisulfide product formed from arsenicin A undergoes reductive desulfurization leading to the formation of a nor-adamantane structure characterized by an As-As bond. The amount of the products obtained depends on the molar equivalents of sulfide salt used, the reaction time, and the reaction temperature [11].

Similarly, by stirring overnight at room temperature compound **7** dissolved in toluene with Na_2_S in water, we obtained the nor-adamantane products **10**–**12** (Figure 2). Unfortunately, the extremely low polarity of these molecules and their limited complete solubility in most solvents prevented a successful HPLC purification, despite several eluent conditions and different stationary phases tried. 

Although NMR analysis highlighted a complex mixture of products, APCI-MS analysis in positive ion mode of the product mixture gave a signal at m/z 452.8, attributable to the [M+H]^+^ ion for the same molecular composition C_4_H_8_As_4_S_3_ of each component, all having three S atoms, as established by high-resolution experiment. The signal at m/z 392.8 obtained by fragmentation experiment on m/z 452.8 was due to the loss of ethanethiol, in line with the typical loss of neutral aldehydes, already observed for natural polyarsenic metabolites and related synthetic structures [1,2,12]. More indications about the molecular structures of the trisulfide products were derived from the study of the reaction mixture NMR spectra (Appendix A). In detail, compound **10** presents two magnetically not equivalent methyl groups and its signals have been quickly assigned. On the contrary, the proposed structures **11** and **12** for the other two products shared magnetically equivalent substituent under *C*_2_ symmetry, resulting in a single set of ^1^H-NMR signals (a doublet and a quartet), correlated to the corresponding ^13^C signals by HSQC experiment. Full assignments were possible through DFT calculations, already successfully applied for similar tetraarsenic cage systems, which are ideal rigid structures for DFT calculation [2,3]. In detail, we adopted the PBE1PBE/aug-cc-pVDZ method since it has been reported to give the best results for ^13^C chemical shift values by the performance evaluation on a series of DFT functionals in combination with different basis sets [16]. A good agreement, specifically for ^1^H-NMR, between computed and experimental values was also obtained in this case (Table 1). The relative abundance 58:28:14 was established for the compounds **10**, **11**, and **12**, respectively, by their integrals in the ^1^H-NMR spectrum of the mixture.

#### 2.1.3. Identification of Natural Arsenicin D and Biogenetic Consideration

Since the identification of the first polyarsenical natural compound arsenicin A in 2006 [1], we have been able to identify a growing number of minor polyarsenical metabolites present in the *E. bargibanti* extract [1,2]. The isolation of these metabolites was carried out through preparative HPLC on the CN column eluting with n-hexane/AcOEt after the removal of sterols and carotenoids from the raw extract. Unfortunately, the full characterization of the metabolite eluted with the first fraction collected at 5.0 min was not achieved, complicated by its extreme scarcity and non-optimal purification. Nevertheless, we are now able to identify this new minor metabolite, named arsenicin D.

High-resolution experiments on the (EI)-MS spectrum (Appendix A) revealed the composition of C_3_H_6_As_4_S_2_ for the molecular ion at *m/z* 405.6796, and a fragment ion at *m/z* 359.6912 deriving from the loss of one molecule of methanethiol, consistently with the recurring fragmentation observed in natural and synthetic molecules belonging to this series of tetraarsenicals. Based on these data, the new minor metabolite was determined as isomeric with arsenicin B [2]. Of note, the relative polarity of all metabolites increases by replacing the S atom with O and by increasing the number of O atoms in the structure, as evident by correlating the corresponding retention times of elution with the molecular composition. In the ^1^H-NMR spectrum of a partially purified fraction, a singlet at 1.49 ppm and two doublets at 2.10 ppm and 3.53 ppm were the signals attributable to this new metabolite. The doublet signals showed a common coupling constant of 12.3 ppm further supported by a 2D-NMR correlation, and were both linked to the same ^13^C signal at 38.9 ppm, as confirmed by the HSQC experiment. These data are in agreement with those reported for one of the symmetric As-S compounds having the same molecular formula as the natural molecule, available by racemic arsenicin A treatment with sodium sulfide in water (Figure 3). Moreover, the structure of this product was confirmed by X-ray analysis [11]. This allowed the identification of structure **13** for arsenicin D.

Arsenicin D was the scarcest metabolite among the polyarsenical series present in the *E. bargibanti* extract, as established by the integrals of the ^1^H-NMR signals of the spectrum acquired for the mixture (Appendix A), indicating the presence of arsenicin A, B, C and D in about a 93:3:3:1 ratio, respectively.

Having now the complete scenario of polyarsenicals isolated from the sponge *E. bargibanti* and improved knowledge of synthetic analogs, some considerations can be made on their production in nature. The production of the nor-adamantane sulfur-cage compounds by the treatment with aqueous sodium sulfide to give **4** (Figure 2) and **13** (Figure 3) as two of the four products obtained [11] may be taken into account for a possible biogenetic hypothesis of these peculiar natural polyarsenicals. Sulfide ions are present in seawater as H_2_S geochemically produced by submarine volcanic activity [17], and biogenic H_2_S may originate from biogenic sources. Additionally, the involvement of cysteine is probable where, after the formation of a stable As-S-bond, dehydroalanine is formed through a beta-elimination. This reactivity is supported by the high affinity of arsenic for sulfur observed in both natural and synthetic processes [2], although the knowledge about thiolation in cells related to As-C bond formation (e.g., methylation) is still scarce [18].

The greater abundance of arsenicin A (evaluated as 93% based on ^1^H-NMR spectroscopy for the fraction containing all polyarsenicals from the workup of *E. bargibanti* extract) in comparison with the other three As-S nor-adamantane metabolites (the remaining 7%) could lead to the assumption of higher stability of the adamantane structure of arsenicin A. Since thermodynamic stability is one of the most relevant driving forces in the production of molecules, we decided to evaluate the energy content of arsenicin B–D structures to be related to the one for arsenicin A. According to DFT calculations at B1B95/6-311+G(3df,2pd) level of theory, arsenicin B, C, and D structures present energy values higher than arsenicin A of 285.96, 227.11 and 285.27 kcal/mol, respectively (Appendix A). Interestingly, also the 3S-analog of arsenicin A, and the fully O-substituted arsenicin B, C and D have energy values higher than the corresponding metabolites (Appendix A), indicating that the most stable structures actually correspond to the isolated natural arsenicins. Therefore, the highest abundance of arsenicin A in the series of polyarsenical metabolites could be explained by the highest thermodynamic stability of this molecule. Anyway, it is worth mentioning that also irreversible biochemical paths leading to less stable isomers (with a small energy gap of a few kcal/mol) are possible (e.g., the natural unsaturated fatty acids as the less stable cis isomers) [19]. This hypothesis could be significant, unless the relative abundance of the majority arsenicin A as compared to arsenicin B-D is due to the presence of the sulfur reactant as the limiting agent.

Lastly, a brief focus on the chirality of these metabolites is notable. The isolated arsenicin A (**1**) showed no optical activity and no Cotton effect in its CD spectrum [1], whereas arsenicin B (**2**) and arsenicin C (**3**) were isolated from the same sponge extract as chiral molecules displaying specific rotations and CD spectra, whose comparison with time-dependent-DFT calculated electronic circular dichroism spectra allowed to establish their absolute configurations [2]. Since arsenicin A has been reported to racemize by a trace acid-catalyzed process [10], we tested in chiral HPLC the fraction containing the polyarsenical mixture from the workup of the sponge extract, revealing the presence of two peaks at 5.3 and 6.3 min (Chiralpak IA, dichloromethane). Similarly, we noticed that the pure enantiomers of each dialkyl analogs **7**–**9** obtained by using the same chiral column, gave fast racemization when dissolved into alcohols or coming into contact with water. Based on this evidence, we can suppose that the use of alcohols and water with possible acid traces in the workup of the sponge extract could have racemized natural arsenicin A, if this was present in an enantio-enriched form. However, it is currently not known whether arsenicin A is produced in pure racemic or enantiomeric form. On the contrary, arsenicin B and C were isolated as enantiomeric forms, in agreement with higher stability towards racemization conditions [11].

### 2.2. Biological Evaluation

Based on the previous results obtained on the antitumor activity evaluation of polyarsenical compounds in leukemia and solid tumors [10,11,12] and the potent antitumor effects exerted by ATO in gliomas in in vitro models [9], we decided to assess the potential use of our synthesized analogs for GBM treatment. In particular, we focused on evaluating their effects on GSCs, which, as previously mentioned, have been associated with all GBM malignant features [14], and are therefore a good therapeutic target in GBM. 

The low polarity and poor solubility of the thio-analogs **10**–**12** hindered their evaluation in biological conditions at suitable concentrations. This behavior is due to the additional methyl groups with respect to the As-S polyarsenical structures reported so far [2,11]. Furthermore, this behavior of **10**–**12** is in line both with the chromatographic profile obtained for the natural arsenicin A-D previously discussed (where replacing O with S atoms considerably decreases the retention time of elution in direct phase HPLC analysis) and with the alkyl chain length causing a remarkable difference in the TLC behavior of compounds **7**–**9**.

Based on these drawbacks observed for **10**–**12**, we focused on evaluating the effect of the alkyl analogs **7**–**9**, each of them available from a highly selective one-step reaction. For all the experiments, ATO was taken as a reference compound.

#### 2.2.1. Alkyl Polyarsenicals Potently and Selectively Inhibit the Growth of GSCs

We started by evaluating the potency of the synthesized alkyl polyarsenical analogs **7**–**9** in inhibiting GSC growth. Indeed, to avoid bias due to GBM inter-tumor heterogeneity, we decided to test the compounds on a panel of nine patient-derived GSC lines, namely COMI, VIPI, GB6, GB7, G144, G166, GB8, GSC#1, and GSC#151. Alongside with GSCs, we also tested these analogs on four non-tumor cell lines to investigate their selectivity. In detail, the non-tumor cell lines tested were MCF10A, from the breast epithelium, ARPE-19, from the retinal pigmented epithelium, Hs68, a fibroblast cell line, and hTERT-HPNE, an immortalized pancreatic ductal cell. The compounds were tested by constructing dose-response curves, and the deriving growth inhibition 50 (GI_50_) value, i.e., the compound concentration causing the 50% inhibition of the cellular growth, was calculated. For this purpose, the cells were treated with a range of drug concentrations in an automatic manner using an acoustic droplet ejection technology-based liquid dispenser and, then, after 48 h treatment, the viability was assessed by performing a vital cell count based on Hoechst 33342 and propidium iodide (PI) DNA stainings. Indeed, the number of viable cells was calculated by subtracting PI-positive cells from the total number of cells, represented by Hoechst 33342 staining (Figure 3a). Figure 3b and Table 2 show the GI_50_ values obtained in the different cell lines tested for each of the four compounds under evaluation.

Although ATO presents GI_50_ values that span the 1–5 µM range on the entire GSC line panel used and also on the non-tumor cell lines tested (except for the Hs68 cell line, for which the GI_50_ value is greater than the highest concentration tested), the alkyl polyarsenicals present GI_50_ values comprised between 0.04–1.5 µM for GSCs and 0.5–7.8 µM for non-tumor cell lines. Remarkably, all three compounds **7**–**9** were more potent than ATO in all the GSC lines tested (Figure 3b and Table 2). Of note, **7**–**9** display different degrees of potency in the GSC lines tested, with **8** and **9** being around fourfold more than **7** (Figure 3b and Appendix A). Indeed, **8** and **9** show 8.7–33.3-fold and 8.7–27-fold greater potency than ATO on the GSC line panel tested (Table 2). These data show that polyarsenical compounds are more potent than ATO not only on GBM cell lines, as previously reported for the antiproliferative activity of synthetic arsenicin A on the U87 glioblastoma cell line (IC_50_ = 0.20 ± 0.09 µM, a value resulting 16.7 more favorable than ATO) [10], but also on GSCs. It is worth mentioning that the GI_50_ value of the currently used drug TMZ on two GSC lines here tested is at least 200-fold higher (GI_50_ values of 95.6 ± 15.2 µM and of 337.7 ± 35.5 µM for COMI and VIPI cells, respectively) [20]. Interestingly, the alkyl polyarsenicals display different GI_50_ values on the different GSCs tested (GI_50_ values range of 0.09–1.46 µM for **7**, 0.05–0.33 µM for **8**, and 0.04–0.28 µM for **9**). The differences observed may be due to the specific mutational background of the different cell lines, and further experiments aimed at understanding the molecular bases of such differential sensitivity could help to unravel these compounds’ mechanisms of action.

To obtain a quantitative parameter of the selectivity of these compounds between GSCs and non-tumor cell lines, we computed a selectivity index (SI), given by the ratio between the GI_50_ value of non-tumor cell lines and the GI_50_ value of GSC lines [21] (Appendix A). As shown in Table 2, although ATO has poor selectivity for GSCs, the alkyl polyarsenicals **7**–**9** display a preferential activity on GSCs compared to non-tumor cell lines. Indeed, the values for the three alkyl analogs were all higher than one, where **8** and **9** showed a SI range of 18.2–124.0 and 12.6–107.3 (Table 2 and Appendix A).

Taken together, these results show that the alkyl polyarsenicals synthesized show both high potency and high selectivity between the GSC and normal cell lines tested and are in favor of the potential applicability of these compounds in therapies that target GSCs. Strikingly, potency and, to a minor extent, also selectivity are enhanced with the elongation of the alkyl chain, with **8** and **9** being around four times more potent and slightly more selective than **7**.

#### 2.2.2. Alkyl Polyarsenicals Display Cytotoxic Effects also on GSC 3D Models

To better characterize the effect of alkyl polyarsenicals on GSC, we used 3D GSC models. Compound **7** was used as a representative for the alkyl analogs **7**–**9**. In detail, we assessed the cytotoxic effect of **7** on GSC COMI gliomaspheres. Indeed, COMI cells show high sensitivity to alkyl polyarsenicals and have been extensively used in our previous works [20,22]. To this aim, 1000 COMI cells were seeded in a low attachment round-bottom plate and allowed to form a gliomasphere for 3–4 days, then the gliomaspheres were treated with different concentrations of **7**. After four days, the gliomaspheres were stained with Calcein AM and SYTOX™ Red Dead Cell Stain dyes (Figure 4a,b), and the viability was measured using the CellTiter-Glo^®^ 3D Cell Viability Assay **(**Figure 4c,d**)**. Compound **7** proved to be more potent than ATO, as previously observed for GSCs grown as adherent cultures. Indeed, **7** can disrupt already formed gliomaspheres at concentrations around 0.3 µM, whereas ATO exhibits the same effect at around 1 µM. These results suggest that **7** maintains its potent antitumor effect on GSCs also in a 3D model, which more closely resembles the tumor environment.

#### 2.2.3. Alkyl Polyarsenicals Induce Apoptosis in GSCs

We next investigated the effect induced by alkyl polyarsenicals on general cell death and cell cycle control pathways. Moreover, for these experiments, **7** was used as a representative for the synthesized alkyl polyarsenical analogs **7**–**9**.

COMI cells were treated either with **7** or ATO at two different concentrations, i.e., around and above the compound’s GI_50_ value. After 48 h of treatment, the protein levels of apoptosis and negative cell cycle regulation markers were analyzed through immunoblotting. In detail, the levels of two apoptosis markers (cleaved caspase 3 and cleaved Poly (ADP-ribose) polymerase (PARP)), of p53, the master regulator of cellular response to stress and DNA damage, and of p21, a cell cycle regulator involved in the G1/S phase transition, were assessed (Figure 5a). Compound **7** is able to induce a strong apoptotic signal in COMI cells at concentrations around its GI_50_ value, as shown by elevated levels of cleaved caspase 3 and PARP. On the other hand, at the concentrations and timepoint tested, ATO seems to have a more cytostatic activity, as indicated by comparably lower levels of apoptotic markers and increased activation of p21. For both **7** and ATO, p53 activation is very slight. p53 lack of activation can be due to several reasons, among them the most common is represented by specific oncogenic mutations in the p53 gene present in the cancer cells. It has been reported that the COMI cell line has a wild-type form of p53 [23]. Indeed, COMI cells treated with camptothecin, a drug known to induce apoptosis through disruption of topoisomerase I activity, massively upregulate p53 (Appendix A). This led us to conclude that neither ATO nor **7** mechanisms of action are strictly dependent on p53 activation, which can be considered an advantage in the treatment of glioblastoma. In fact, p53 alterations, both direct and indirect, are often found in this type of tumor and are, at least partially, responsible for treatment resistance [24]. To evaluate the dose dependency of apoptotic induction exerted by **7**, COMI cells were treated with different concentrations of **7** (0.05, 0.1, 0.2, and 0.4 μM). As depicted in Figure 5b, **7** can induce apoptosis in a dose-dependent manner, peaking at a concentration of 0.2 μM.

To further investigate the timing of apoptosis induction exerted by **7**, COMI cells were treated with 0.2 μM of **7**, and immunoblot analyses of apoptosis and cell cycle regulation markers were performed at different time points (10 h, 24 h, 36 h, 48 h) (Figure 6). In COMI cells treated with 0.2 μM of **7**, very little apoptosis was detected in the first 24 h, whereas a minor increase in p21 protein levels was observed. Subsequently, at 36 h, apoptotic markers started to increase and reach their maximum at 48 h, whereas instead p21 levels decreased slightly.

Taken together, these results show that **7** can strongly induce apoptosis in COMI cells in a dose-dependent manner at concentrations around its GI_50_ value. The apoptotic activity observed for **7** starts at 36 h and peaks at 48 h.

#### 2.2.4. Alkyl Polyarsenicals Maintain their Potency under Hypoxic Conditions 

Hypoxia has been extensively studied in the context of glioblastoma and GSCs. Indeed, GSCs are known to occupy hypoxic niches of the tumor, which favor stemness maintenance and drug resistance [25,26]. Moreover, among arsenic species’ mechanisms of action, the induction of oxidative stress via impairment of the electron transport chain and oxidative phosphorylation (OXPHOS) in the mitochondria is one of the most studied [27]. Although it has been shown that as little as 1% oxygen is sufficient for effective OXPHOS in GSCs [28], a reduction in oxygen concentrations could reduce cellular aerobic metabolism, potentially leading to a decrease in the sensitivity of GSCs to ATO and alkyl polyarsenicals, even though the mechanism of action of the latter has not been elucidated yet. In this context, the effect of alkyl polyarsenicals **7**–**9** on cell growth inhibition was evaluated on a GSC line cultured under hypoxic conditions (1% oxygen). In detail, COMI cells were cultured either under normal oxygen or under hypoxic conditions (21% and 1% oxygen, respectively) and treated either with ATO or one of the compounds **7**–**9** at different concentrations for 48 h. To assess the effects caused by the metabolic and transcriptomic changes induced by growth in hypoxic conditions on the cellular response to polyarsenical compound treatments, two different experimental settings were used, i.e., exposure to 1% oxygen only during the compound treatment (48 h) or for a week prior to drug treatment. Figure 7 summarizes the GI_50_ values calculated for the different conditions and compounds tested. The numeric GI_50_ values can be found in Appendix A. At 1% oxygen, the viability of untreated cells was not affected, and the cells still responded to polyarsenical compound treatment in a dose-dependent manner, supporting their efficacy on GSCs under hypoxic conditions, such as those present in hypoxic niches of the tumor. Interestingly, also ATO potency is not impacted by low oxygen conditions. This can be explained by the fact that cells under hypoxic conditions still have effective OXPHOS [28], but also because induction of oxidative stress is not the only mechanism of action through which ATO induces cell death [9].

### 2.3. Physicochemical Properties and ADME Prediction

For the potential translation to the clinics, a small molecule has to present a suitable drug-likeness profile by meeting specific chemical physical criteria. Detailed parameters can be predicted using the SwissADME tool [29]. Besides respecting Lipinski’s rule, alkyl analogs **7**–**9**, As-S noradamantane structures **10**–**12,** and arsenicin A (**1**), which was taken as a reference, show proper physicochemical properties, visualized in the bioavailability radar reported in Appendix A. In detail, the lipophilicity trend increases moving from arsenicin A toward the alkyl analogs **7**–**9**, with higher values correlating with longer alkyl chains, and in turn toward the dimethyl As-S nor-adamantane **10**–**12**, which are not distinguishable from each other.

The topological polar surface area (TPSA) parameter, which can be deduced based on the sum of the contributions of the different atoms and moieties present in a specific molecule, is a valuable descriptor to estimate the capability of that given molecule to cross biological barriers [30]. The predicted values of this parameter are 27.69 Å^2^ for arsenicin A (**1**), 36.92 Å^2^ for **7**–**9,** and 75.90 Å^2^ for **10**–**12**. Lipophilicity and TPSA correlation is visualized in the Brain Or IntestinaL EstimateD permeation method (BOILED)-Egg diagram [30]. The results obtained (Appendix A) highlight the better blood–brain barrier (BBB) permeation capability of alkyl and As-S analogs in comparison with arsenicin A (**1**). This is notable if correlated to ATO, whose use in clinical trials for glioma treatment is greatly limited by its low permeability of BBB [9]. Glioblastoma-induced alterations can increase BBB permeability, thus allowing a molecule to distribute more into the glioblastoma bulk than in the normal brain parenchyma, as reported by Carmignani et al. for ATO [31]. Moreover, the use of nanocarriers can also favor ATO-targeted delivery, although at the present further studies are needed [8]. However, the use of molecules potentially able to cross the BBB, as the alkyl polyarsenicals **7**–**9**, is more advantageous. Additionally, gastrointestinal absorption was estimated to be high for all molecules considered in our work, which also have a very good bioavailability score (0.55), a parameter that should not be lower than 0.25 [30]. However, due to the peculiar structure of these polyarsenicals, care should be taken in the use of these predictive models, which have been developed considering more classic organic molecules. We compared the SwissADME data with those obtained using Molsoft L.L.C. [32] and Molinspiration [33], the latter indicated as able to process most organometallic molecules. Although expressed in different forms using the three tools (Appendix A), the trend for lipophilicity is similar, observing an obvious increase in moving from arsenicin A to dimethyl, diethyl, and dipropyl analogs **7**–**9**, respectively; the PSA parameters do not distinguish alkylated polyarsenicals from each other, and the BBB permeation results were favorable for compounds with ethyl and propyl chains.

Taken together, these results show that, in addition to potent and selective GSC growth inhibition, the alkyl polyarsenicals **7**–**9**, and especially **8** and **9,** possess a good drug-likeness profile and show the advantage to potentially cross the BBB without the need for specific delivery systems.

## 3. Materials and Methods

### 3.1. Chemistry

#### 3.1.1. General Experimental Procedures

All chemicals and reagents were purchased from Sigma Aldrich (Taufkirchen, Germany). CAUTION: due to the hazard statement for arsenic trioxide (H350), a reduced reaction scale and all the safety conditions in the workup and manipulation were adopted. Carboxylic acids and anhydrides used are characterized by a strong and unpleasant smell. Yields are referred to as purified compounds.

Thin layer chromatography (TLC) was performed using Merck silica gel F_254_ using short-wave UV light as the visualizing agent, and KMnO_4_ as developing agents upon heating. Preparative thin layer chromatography (PLC) was performed using 20 × 20 cm Merck Kieselgel 60 F_254_ 0.5/2-mm plates. Column chromatography was performed using Merck Si 45–60 µm as stationary phase. The HPLC chromatograms were obtained by a reversed-phase RP-HPLC system using an Agilent 1200 high-performance liquid chromatography (HPLC) system equipped with an autosampler, a binary pump, a diode array detector (Agilent Technologies Waldbronn, Germany) and Chiralpak IA column (5 μm, 250 mm × 4.6 mm), detection at 254 nm, flow 1 mL/min. Infrared spectra were recorded on the film from the evaporation of a CH_2_Cl_2_ solution, by using an FT-IR Tensor 27 Bruker spectrometer equipped with an Attenuated Transmitter Reflection (ATR) device at 1 cm^−1^ resolution in the absorption region Δν 4000−1000 cm^−1^. Spectral processing was made using the Opus software package (version 7.5.18). NMR spectra were recorded on a Bruker-Avance 400 spectrometer using a 5 mm BBI probe ^1^H at 400 MHz and ^13^C at 100 MHz and calibrated using residual undeuterated solvent for CDCl_3_ (relative to δ_H_ 7.25 ppm and δ_C_ 77.0 ppm, respectively) with chemical shift values in ppm and *J* values in Hz. The following abbreviations were used to explain multiplicities: s = singlet, d = doublet, t = triplet, q = quartet, m = multiplet, and br = broad. NMR data were analyzed using BrukerTopspin software 3.6.1 version. Assignments were by ^1^H,^1^H-COSY and heteronuclear single quantum correlation (HSQC) experiments. APCI-MS and tandem (MS/MS)^n^ were taken through a Bruker Esquire-LC mass spectrometer equipped with an atmospheric pressure chemical ionization ion (APCI) source used in positive ion mode. The sample was injected into the source from a methanolic solution; high resolution (HR)-APCI experiments were performed using a Water Xevo G2−XS QTof spectrometer; electron ionization (EI)-MS (*m/z*, rel.%) and HR-EIMS spectra were taken by a Kratos-MS80 mass spectrometer equipped with a home-built acquisition system. The IUPAC nomenclature of the compounds was provided by using MarvinSketch 22.11 (obtained from Chemaxon https://chemaxon.com/)(accessed on 26 January 2023).

#### 3.1.2. Synthesis of Alkyl Analogs **7**–**9**

##### 9,10-Dimethyl-2,4,6,8-tetraoxa-1,3,5,7-tetraarsatricyclo[3.3.1.1^3,7^]decane (**7**)

The compound was obtained by reaction of arsenic trioxide with propionic acid, propionic anhydride, and K_2_CO_3_ as previously reported [1]. TLC: dichloromethane/n-hexane 70:30 *v/v*, R_f_ = 0.29. ^1^H-NMR (400 MHz, CDCl_3_) δ: 1.59 (d, *J* = 7.8 Hz, 3H), 1.40 (q, *J* = 7.8 Hz, 1H); ^13^C-NMR (100 MHz, CDCl_3_) δ: 32.3, 10.3. APCI(+)MS: *m/z* 420.9 [M+H]^+^, APCI (+) MS/MS (420.9): *m/z* 402.8, 376.9, 358.8, 348.9, 332.9. Both HRMS and experimental/simulated IR data were already reported in Mancini et al. [1]. The chromatogram from the chiral HPLC analysis (dichloromethane/hexane 50:50 *v/v*) gave two peaks of equal intensity at t_R_ = 5.9 min and 7.4 min.

##### 9,10-Diethyl-2,4,6,8-tetraoxa-1,3,5,7-tetraarsatricyclo[3.3.1.1^3,7^]decane (**8**)

A mixture of arsenic(III) oxide (160 mg, 0.8 mmol), anhydrous K_2_CO_3_ (113 mg, 1 eq.), butyric acid (0.3 mL, 4 eq.), and butyric anhydride (1.1 mL, 8 eq.) was stirred for 3 h at 180 °C under N_2_ atmosphere. Water (2 mL) was then added and the mixture was heated for 1.5 h at 90 °C. Additional water was added and the mixture was extracted with dichloromethane (3 × 10 mL). The reaction progress was monitored by TLC (in dichloromethane), observing the appearance of a spot at R_f_ = 0.9 (254 nm). The white solid, unreacted arsenic trioxide, was removed and the organic layer was dried over anhydrous Na_2_SO_4_ and concentrated under reduced pressure. The residues were purified by PLC on silica gel (100% dichloromethane). Whitish powder; yield: 17% (30 mg). TLC: dichloromethane/n-hexane 70:30 *v/v*, R_f_ = 0.47. FT-IR (cm^−1^): 2954 (w), 2923 (w), 2843 (w), 1451 (m), 1043 (m), 795 (vs), 723 (s), 456 (s). ^1^H-NMR (400 MHz, CDCl_3_) δ: 2.13 (dq, *J* = 7.5 Hz, *J* = 7.5 Hz, 2H), 1.33 (t, *J* = 7.5 Hz, 1H), 1.25 (t, *J* = 7.5 Hz, 3H); ^13^C-NMR (100 MHz, CDCl_3_) δ: 41.3, 20.1, 14.2. (Appendix A) APCI(+)MS: *m/z* 448.8 [M+H]^+^, APCI (+) MS/MS (448.8): *m/z* 431.0, 406.9, 388.8, 370.8, 364.8, 320.0. The chromatogram from the chiral HPLC analysis (dichloromethane/hexane 35:65 *v/v*) gave two peaks of equal intensity at t_R_ = 8.4 min and 10.6 min.

##### 9,10-Dipropyl-2,4,6,8-tetraoxa-1,3,5,7-tetraarsatricyclo[3.3.1.1^3,7^]decane (**9**)

A mixture of arsenic(III) oxide (86 mg, 0.43 mmol), anhydrous K_2_CO_3_ (45 mg, 1 eq.), valeric acid (0.4 mL, 4 eq.), and valeric anhydride (0.8 mL, 8 eq.) was stirred for 3 h at 200 °C under N_2_ atmosphere. Water (2 mL) was then added and the mixture was heated for 1.5 h at 90 °C. Additional water was added and the mixture was extracted with dichloromethane (3 × 10 mL). The reaction progress was monitored by TLC (eluting in dichloromethane), observing the appearance of a spot at R_f_ = 0.9 (254 nm). The white solid, unreacted As_2_O_3_, was removed and the organic layer was dried over anhydrous Na_2_SO_4_ and concentrated under reduced. The residues were purified by PLC on silica gel (100% dichloromethane). Whitish powder; yield: 15% (15 mg). TLC: dichloromethane/n-hexane 70:30 *v/v*, R_f_ = 0.59. FT-IR (cm^−1^): 2953 (m), 2916 (m), 2870 (m), 1461 (m), 1063 (m), 789 (vs), 717 (s), 465 (s). ^1^H-NMR (400 MHz, CDCl_3_) δ: 2.05 (dt, *J* = 7.7 Hz, *J* = 8.0 Hz, 2H), 1.69–1.58 (m, 2H), 1.40 (t, *J* = 7.5 Hz, 1H), 1.00 (t, *J* = 7.2 Hz, 3H); ^13^C-NMR (100 MHz, CDCl_3_) δ: 39.1, 28.9, 22.6, 14.0. (Appendix A) APCI(+)MS: *m/z* 476.9 [M+H]^+^, APCI (+) MS/MS (476.9): *m/z* 458.8, 420.9, 402.8, 364.8, 348.9. HR-APCI(+)MS: *m/z* 476.7985 [M+H]^+^, calcd. for C_8_H_17_As_4_O_4_: 476.798519. The chromatogram from the chiral HPLC analysis (dichloromethane/hexane 35:65 *v/v*) gave two peaks of equal intensity at t_R_ = 9.6 min and 11.7 min.

#### 3.1.3. Synthesis of Sulfur Analogs **10**–**12**

A solution of Na_2_S∙9H_2_O (41 mg, 0.17 mmol, 3 eq.) in water (3.5 mL) was added to a solution of **7** (24 mg) in toluene (20 mL) under a nitrogen atmosphere. The mixture was stirred at room temperature overnight and then extracted with toluene (3 × 10 mL). The organic layer was dried over Na_2_SO_4_ and concentrated under reduced pressure. The whitish residue was eluted with dichloromethane through a short pad of silica gel (Merck Si 40−63 μm) giving a mixture of products **10**–**12**, with a global recovery = 45%. TLC: dichloromethane/n-hexane 70:30 *v/v*, R_f_ = 0.91. APCI(+)MS: *m/z* 452.8 [M+H]^+^, APCI (+) MS/MS (452.8): *m/z* 392.8, 256.9; HR-APCI(+)MS: *m/z* 452.6725 [M+H]^+^, calcd. for C_4_H_9_As_4_S_3_: 452.672477. FT-IR (cm^−1^): 2952 (w), 2864 (w), 1727 (s), 1433 (s), 1368 (m), 1002 (s), 790 (s), 621 (s), 489 (m), 468 (m).

The diastereoisomeric mixture was evaluated as 58:28:14 for **10**/**11**/**12**, respectively by the integrals of ^1^H-NMR signals (Appendix A).

(4R*, 8S*)-4,8-Dimethyl-2,6,9-trithia-1,3,5,7-tetraarsatricyclo[3.3.1.0^3,7^]nonane (**10**). ^1^H-NMR (400 MHz, CDCl_3_) δ: 3.97 (q, *J* = 7.9 Hz, 1H), 2.51 (q, *J* = 7.9 Hz, 1H), 2.12 (d, *J* = 7.9 Hz, 3H), 1.11 (d, *J* = 7.9 Hz, 3H); ^13^C-NMR (100 MHz, CDCl_3_) δ: 48.8, 43.0, 20.0, 13.3. HSQC correlations: 3.97/48.8, 2.12/13.3, 1.11/20.0.

(4R*, 8R*)-4,8-Dimethyl-2,6,9-trithia-1,3,5,7-tetraarsatricyclo[3.3.1.0^3,7^]nonane (**11**). ^1^H-NMR (400 MHz, CDCl_3_) δ: 3.76 (q, *J* = 7.2 Hz, 1H), 1.07 (d, *J* = 7.2 Hz, 3H); ^13^C-NMR (100 MHz, CDCl_3_) δ: 48.8, 19.8. HSQC correlations: 3.76/48.8, 1.07/19.8.

(4S*)-4,8-Dimethyl-2,6,9-trithia-1,3,5,7-tetraarsatricyclo[3.3.1.^3,7^]nonane (**12**). ^1^H-NMR (400 MHz, CDCl_3_) δ: 2.61 (q, *J* = 7.2 Hz, 1H), 2.21 (d, *J* = 7.2 Hz, 3H); ^13^C-NMR (100 MHz, CDCl_3_) δ: 42.5, 12.9. HSQC correlations: 2.61/42.5, 2.21/12.9.

#### 3.1.4. Identification of Arsenicin D

The workup corresponds to the procedure previously reported in detail for arsenicin A-C [1,2]. Briefly, the sponge was collected along the northeastern coast of New Caledonia, then in sequence freeze-dried, extracted (EtOH), evaporated, and CH_2_Cl_2_/H_2_O partitioned. The CH_2_Cl_2_ extract was partitioned by flash chromatography (Si-60, n-hexane/AcOEt) and the combined fractions 6–15 were subjected to reversed-phase flash chromatography (RP-18, MeCN/H_2_O) to separate the UV spots corresponding to arsenicals from sterols and carotenoids. After evaporating acetonitrile, the remaining aqueous phase was extracted with AcOEt. The residue from evaporation was subjected to preparative HPLC (Lichrosorb CN, 7 mm, UV detection at λ 254 nm) with n-hexane/AcOEt 96:4 *v/v*. The chromatographic profile shows peaks for increasing retention times, corresponding to the minor arsenicin D (t_R_ 5.0 min), arsenicin B, arsenicin C, and the major arsenicin A. The fraction collected at 5 min turned out to be non-pure and the scarce quantity prevented the isolation of pure arsenicin D.

##### Arsenicin D (2,6-dithia-1,3,5,7-tetraarsatricyclo[3.3.1.0^3,7^]nonane, **13**)

^1^H-NMR (400 MHz, CDCl_3_) δ: 3.53 (d, *J* = 12,3 Hz, 2H), 2.10 (d, *J* = 12.3 Hz, 2H), 1.49 (s, 2H), (Appendix A); ^1^H,^1^H-COSY correlation: 2.10/3.53; HSQC correlations: 2.10/38.9, 3.53/38.9. (EI)-MS: *m/z* (%) 405.8 [M^+^, 68]^+^, 359.8 (32), 316.8 (23), 181.9 (9), 106.9 (25), (Appendix A); HREI-MS: *m/z* 405.67960 [M^+**.**^], calcd. for C_3_H_6_As_4_S_2_: 405.67748; *m/z* 359.6912 [M^+**.**^−H_2_CS], calcd. for C_2_H_4_As_4_S: 359.6899.

### 3.2. Computational Details

DFT calculation was performed in the gas phase and in chloroform by using the Conductor-like Polarized Continuum Model (C-PCM) [34]. Calculations were carried out on a PC running at 3.4 GHz on an AMD Ryzen 9 5950X 16 core (32 threads) processor with 32 GB RAM and 1 TB hard disk with Windows 10 Home 64-bit as an operating system. The structures of the compounds were built with GaussView 6.0 and the Gaussian 03W revision E.01 program [35] was used in the geometry optimization at a density functional theory (DFT) level of theory. The optimized geometry and IR calculation were obtained by using the RFO step, integral precision = superfine grid and type convergence criteria, and invoking gradient employing 6-311+G(3df,2pd) basis set for all atoms. The electronic correlation functional B1B95, where the gradient-corrected DFT with Becke hybrid functional B1 [36] for the exchange part and the B95 for correlation function [37,38,39] was utilized. The optimized structural parameters were taken in the vibrational energy calculations at the DFT levels to characterize all stationary points as minima. No imaginary wave number modes were obtained for the optimized structure, proving that a local minimum on the potential energy surface was actually found. NMR simulation was carried out employing aug-cc-pVDZ basis set with hybrid generalized gradient approximation (H-GGA) functional PBE1PBE aka PBE0 [40]. Magnetic properties were calculated with GIAO schemes [41,42].

### 3.3. Biological Evaluation

#### 3.3.1. Cell Culture

Human glioblastoma stem cell lines COMI and VIPI were kind gifts from Antonio Daga (San Martino University Hospital of Genoa, Italy) [23,43,44], and were cultured in DMEM/F-12 and Neurobasal in a 1:1 ratio (Thermo Fisher Scientific, Waltham, MA, United States), supplemented with 2 mM GlutaMAX (Thermo Fisher Scientific), 100 U/mL penicillin G (Sigma Aldrich, Taufkirchen, Germany), 1% B27 supplement (Thermo Fisher Scientific, Waltham, United States), 20 ng/mL recombinant human epidermal growth factor (EGF) (R&D Systems, Minneapolis, United States), 10 ng/mL recombinant human fibroblast growth factor-2 (bFGF) (R&D Systems), and 2 µg/mL heparin (Sigma Aldrich, Taufkirchen, Germany). Human glioblastoma stem cell lines GB6, GB7, G144, G166, and GB8 were kind gifts from Luciano Conti (Department CIBIO, University of Trento) [45,46,47], and GSC#1, and GSC#151 were kind gifts from Lucia Ricci-Vitiani (Istituto Superiore di Sanità) [48,49]. These human glioblastoma cell lines were cultured in Euromed-N media (Euroclone), supplemented with 2 mM GlutaMAX (Thermo Fisher Scientific, Waltham, United States), 100 U/mL Penicillin G (Sigma Aldrich, Taufkirchen, Germany), 2% B27 supplement (Thermo Fisher Scientific, Waltham, United States), 1% N2 supplement (Thermo Fisher Scientific, Waltham, United States), 20 ng/mL EGF (R&D Systems, Minneapolis, United States), 20 ng/mL bFGF (R&D Systems, Minneapolis, United States). All glioblastoma stem cell lines were cultured as adherent cultures on laminin-precoated flasks, except for GSC#1 and GSC#151 which were cultured on vitronectin-precoated flasks. Culture flasks precoated with laminin were incubated for 3 h at 37 °C or overnight at 4 °C prior to use. Culture flasks precoated with vitronectin were incubated for 1 h at 37 °C prior to use.

For the experiments on gliomaspheres, COMI cells were cultured in suspension without the laminin coating.

Human breast epithelium cell line MCF10A was obtained from ATCC (CRL-10317™) and was cultured in DMEM-F12 (Thermo Fisher Scientific, Waltham, United States), supplemented with 5% of horse serum, 2 mM GlutaMAX (Thermo Fisher Scientific), 100 U/mL penicillin G (Sigma Aldrich, Taufkirchen, Germany), 20 ng/mL recombinant human epidermal growth factor (EGF) (R&D Systems, Minneapolis, United States), 0.5 µg/mL of hydrocortisone, 100 ng/mL of cholera toxin, and 10 µg/mL of human recombinant insulin. Human pancreatic duct cell line hTERT-HPNE was obtained from ATCC (CRL-4023™) and was cultured in DMEM and M3 in a 3:1 ratio, supplemented with 10% fetal bovine serum, 2 mM GlutaMAX (Thermo Fisher Scientific, Waltham, United States), 100 U/mL Penicillin G (Sigma Aldrich, Taufkirchen, Germany), and 25 µg/mL of gentamycin, 10 ng/mL recombinant human epidermal growth factor (EGF) (R&D Systems, Minneapolis, United States). Human retinal pigmented epithelium cell line ARPE-19 was obtained from ATCC (CRL-2302™) and was cultured in DMEM-F12 (Thermo Fisher Scientific, Waltham, United States), supplemented with 5% of fetal bovine serum, 2 mM GlutaMAX (Thermo Fisher Scientific, Waltham, United States), and 100 U/mL Penicillin G (Sigma Aldrich, Taufkirchen, Germany). Human foreskin fibroblast cell line Hs68 was obtained from ATCC (CRL-1635™) and was cultured in DMEM, supplemented with 10% of fetal bovine serum, 2 mM GlutaMAX (Thermo Fisher Scientific, Waltham, United States), and 100 U/mL Penicillin G (Sigma Aldrich, Taufkirchen, Germany).

All cell cultures were kept at 37 °C and 5% CO_2_. For hypoxia experiments, cells were grown in a hypoxic incubator at 37 °C, 5% CO_2_, and 1% O_2_.

#### 3.3.2. Preparation of Solutions of the Compounds to Be Tested

ATO solutions for cell growth inhibition testing were prepared by solubilizing ATO powder in the calculated amount of sterile water to obtain a 10 mM solution. Solutions were stored at −20 °C and heated at 80 °C for at least 20 min before use. Each solution of **7**–**9** was prepared by solubilizing the powdered compound in the calculated amount of cell culture-grade DMSO to obtain a 10 mM solution. Solutions were stored at −20 °C and heated at 80 °C for at least 20 min before use. For compounds **10**–**12**, which presented poor DMSO solubility, different concentrations were tried. The 10 mM and 1 mM solutions appeared cloudy and with visible precipitates even after thorough mixing and heating at 80 °C. The maximum DMSO concentration able to give a clear solution was 0.5 mM, but this was unsuitable for preliminary in vitro tests since the concentration of DMSO in the cell medium obtained using this solution as stock compound exceeded 1%, which is usually the threshold.

#### 3.3.3. Treatments Performed Using Echo 650 and Data Analysis

Cells grown in culture flasks were detached using either accutase or trypsin. Cells were then counted using a trypan blue staining and an automated cell counter and seeded in a SpectraPlate-384 TC (PerkinElmer) and incubated for 24 h at 37 °C, 5% CO_2_. The optimal seeding numbers for each cell line are reported in Appendix A. Cells were treated with different concentrations of either ATO, **7**, **8**, or **9** using BeckmanCoulter Echo 650 Dose-Response protocol. Each treatment was performed in technical triplicate. After the treatment, cells were incubated for 48 h. Then, cells were stained with Hoechst 33342 (1 µg/mL, Thermo Fisher Scientific, Waltham, United States, cat. H1399) and propidium iodide (PI, 1 µg/mL, Sigma Aldrich, Taufkirchen, Germany cat. P4170) and incubated at 37 °C for 1 h in the dark. Fluorescence images were acquired using the MolecularDevices ImageXpress Micro Confocal High Content Imaging System and analyzed using a custom module of MetaXpress analysis software, version 6.7.1.157. The number of viable cells was computed by subtracting PI-positive cells from the total number of cells, represented by Hoechst 33342 staining. Live cell numbers were normalized on their respective non-treated control. Data were then analyzed using the DRC node from R in the Knime platform to obtain dose-response curves and calculate GI_50_ values.

#### 3.3.4. Treatment of Gliomaspheres

COMI cells grown as gliomaspheres were mechanically dissociated to single cells, counted, and plated at a density of 1000 cells per well in Ultra-Low attachment round bottom 96-well plates (Costar). Cells were incubated for 3–4 days to allow the formation of a gliomasphere. Then, cells were treated with different concentrations of **7** or ATO and incubated for additional 4 days. Each treatment was performed in a technical quadruplicate. At the end of the treatment, spheres were stained with 1 µM Calcein AM (Thermo Fisher Scientific, Waltham, United States, cat. C3100MP) and 0.02 µM SYTOX™ Red Dead Cell Stain (Thermo Fisher Scientific, Waltham, United States, cat. S34859) for 1 h at 37 °C in the dark and images were acquired with the MolecularDevices ImageXpress Micro Confocal High Content Imaging System. Afterward, viability was determined using the CellTiter-Glo^®^ 3D Luminescent Cell Viability Assay (Promega, Madison, WI, United States, cat. G7570) as per the manufacturer’s instructions. Viability was calculated as the percentage of the non-treated control.

#### 3.3.5. Western Blot

COMI cells were seeded in 6-well culture plates at a density of 120,000 cells per well and incubated for 24 h at 37 °C, 5% CO_2_. Cells were then treated with the required concentration of compound for the required amount of time, depending on the experimental setting. At the end of the treatment, cells were harvested and centrifuged, then washed in PBS and resuspended in RIPA lysis buffer (Merck, Rahway, NJ, USA) supplemented with protease inhibitors and phosphatase inhibitors. Protein concentrations were quantified with PierceTM BCA Protein Assay Kit (Thermo Fisher Scientific, Waltham, United States, cat. 23227) as per the manufacturer’s protocol. For each sample, equal amounts of protein (20–30 µg) were loaded on SDS-PAGE and separated at 130 V for 1.5 h. Proteins were then transferred to a PVDF membrane at 350 mA for 1.5 h. Membranes were blocked in 5% milk prepared in TBS-Tween 0.2%, then probed with anti-PARP (Cell Signaling, Technology, Danvers, MA, United States, cat. 9542), anti-caspase 3 (Cell Signaling, Technology, Danvers, United States, cat. 9662), anti-p53 (Santa Cruz, Biotechnology, Dallas, TX, United States, cat. sc-126), anti-p21 (Abcam, Cambridge, UK, cat. ab109520), anti-β-tubulin (Santa Cruz, Biotechnology, Dallas, United States, cat. sc-53140), anti-GAPDH (Santa Cruz, Biotechnology, Dallas, United States, cat. sc-32233), and HRP-conjugated antibodies (Invitrogen, Waltham, MA, United States). Membranes were incubated with primary antibodies for 1 h at room temperature, or overnight at 4 °C. Secondary antibodies were added after three washes in TBS-Tween 0.2% and incubated for 1 h at room temperature. The signal was detected after another three washes in TBS-Tween 0.2% by using Amersham ECL Prime or Select Western Blotting Detection Reagent (GE Healthcare Life Sciences, Chicago, IL, United States) and ChemiDoc Imaging System (Bio-Rad, Hercules, CA, United States). Data were analyzed using ImageLab software 3.0.

## 4. Conclusions

The series of polyarsenical adamantane and nor-adamantane compounds inspired by the peculiar structures of the marine metabolites arsenicin A-C has been here enriched by synthesizing the dialkyl analogs **7**–**9**, which were efficiently obtained by a selective one-pot reaction, and the new dimethyl thio-compounds **10**–**12**, which have been structurally characterized with the support of simulated NMR spectra. In addition, the minor metabolite arsenicin D has also been identified in the mixture of the metabolites isolated from the *E. bargibanti* extract and some considerations on the arsenicin A-D biogenetic production are provided.

In light of the promising antitumor activity previously observed for arsenicin A-C analogs, **7**–**9** have been evaluated for their potential use in GBM treatment. Compound **7** had previously been evaluated as an efficient inhibitor of the growth of a large series of tumor cells, but not including glioblastoma, or tumor stem cells. In detail, these compounds have been characterized for their capability to affect GSCs, which are emerging as an interesting therapeutic target. We showed that **7**–**9** potently and selectively inhibit GSC growth, with GI_50_ values and selectivity indexes better than those of ATO, especially for compounds **8** and **9**. Polyarsenical **7**, taken as representative of the dialkyl analogs, maintained its effects on a GSC 3D model and acted by inducing apoptosis. We also proved that **7**–**9** retain their potency on GSCs in hypoxic conditions, such as those present in hypoxic niches of the tumor. Finally, according to physicochemical parameters evaluation and ADME prediction, **7**–**9** present good properties and good BBB permeation. In conclusion, dialkyl polyarsenicals, and in particular compounds **8** and **9**, possess interesting antitumor properties and are definitely worthy of further evaluation.

## Data Availability

Not applicable.

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
