# Peer review of "Expanding the Chemical Space of Arsenicin A-C Related Polyarsenicals and Evaluation of Some Analogs as Inhibitors of Glioblastoma Stem Cell Growth"

_marinedrugs, 2023, doi:10.3390/md21030186_

Round 1
Reviewer 1 Report
After their spectacular discovery of arsenicin A as a natural product, Mancini and coworkers report some less abundant sulfur containing congeners with similar general structures. The paper is a great read and I am happy to have read it even prior to publication. This is yet another great piece of work and definitely deserves publication, particularly since the antiproliferative properties of the products have also been evaluated.
My queries, comments and corrections can be found in the attached commented pdf file. Essentially, some more HRMS data would be helpful to further support the structures presented. Could IR and Raman spectra be calculated and compared with actual recordings?
After some revisions, I strongly support publication of the manuscript in Marine Drugs.

Author Response
From the authors:
We are very grateful to the reviewer for the time devoted to this revision, as well as for the valuable feedback and constructive comments on the manuscript.The changes are currently highlighted within the new version of the manuscript. Please find below, point-by-point our responses to the specific comments and suggestions.
After their spectacular discovery of arsenicin A as a natural product, Mancini and coworkers report some less abundant sulfur containing congeners with similar general structures. The paper is a great read and I am happy to have read it even prior to publication. This is yet another great piece of work and definitely deserves publication, particularly since the antiproliferative properties of the products have also been evaluated.
My queries, comments and corrections can be found in the attached commented pdf file. Essentially, some more HRMS data would be helpful to further support the structures presented.
Answer from the authors: Data from HR-APCIMS experiments have been introduced for the new compound 9 and the isomeric mixture of new S-analogues 10-12, as the indication of the instrument used.
Could IR and Raman spectra be calculated and compared with actual recordings?
Answer from the authors:
As reported in the text, based on the experience and reference data acquired in the study of similar structures, since the first report on the arsenicin A, the structures of alkyl polyarsenicals 8 and 9 was elucidated by NMR and MS analyses, obtaining data in line with those of the extensively studied lower homologues. As a response to this comment, we have now recorded the FT-IR spectra (adding these results in the structural description) and compared them with the values of the corresponding simulated spectra, observing a very good agreement. The values and assignments are shown in the new Table S1. The following sentence has been added to the text: “A nice agreement was observed between experimental and DFT-calculated IR spectra of both compounds 8 and 9 (Table S1). Furthermore, the low-frequency active transitions assigned to As-O stretching (797 and 796 cm-1 for 8 and 9, resp.) and to As-C stretching (727 and 731 cm-1 for 8 and 9, resp.) agreed with the values previously reported for dimethyl analogue 7 [1].”
After some revisions, I strongly support publication of the manuscript in Marine Drugs.
Line 21 : of polyarsenicals related to arsenicin A by synthesis
Line 23 : the scarcity of which
Line 29 : both under
Line 72 : Synthetic arsenicin A and related polyarsenicals (Figure 2)
Line 92: Synthetic arsenicin A and related polyarsenicals (Figure 2)
Lines 118-127 : numbering in bold : done
Schemes 1 and 2: done
Line 176: through DFT calculations
- Comment at line 229: there is also biogenic H2S and it has been recognized as a gaseous transmitter even in mammals, just like NO and CO. Another source could be cysteine, where, after formation of a stable As-S-bond, dehydroalanine is formed through beta-elimination.
Answer from the authors:
We thank the reviewer for the clarification; indeed we had thought about the involvement of cysteine, but then we hadn't reported it. The text has been changed as follows:
“Sulfide ions are present in seawater, as H2S geochemically produced by submarine volcanic activity [17], and biogenic H2S may originate from biogenic sources. Additionally, the involvement of cysteine is probable where, after formation of a stable As-S-bond, dehydroalanine is formed through a beta-elimination.”
Line 234 : based on 1H-NMR spectroscopy
Line 237 : could lead to the assumption of a higher
- Comment at line 237: does this relate to kinetic stability? Thermodynamically, it can't be true as arsenicin A would not react with sulfur if it was thermodynamically more stable than the sulfur-containing arsenicins.
Answer from the authors
There are reactions of thermodynamically stable compounds and yet kinetically labile (reactive), and vice-versa. In the case of an endothermic reaction, the products are less stable than the reactants.
The high affinity of As for S (widely reported. i.e. reference 2) could rule the reactivity of these polyarsenicals to give thio-derivatives, from a reaction we see occurring at room temperature.
- More interesting than absolute values would be relative values for the reaction of arsenicin A with H2S to H2O and the sulfur-containing arsenicin. What would be the outcome here?
Answer from the authors
The appreciated suggestion has been taken into consideration: the relative values of energies calculated for the indicated structures have been reproduced in Table S2, and in the text it has been specified:
“Interestingly, also the 3S-analogue of arsenincin A, and the fully O-substituted arsenicin B, C and D have energy values higher than the corresponding metabolites (Table S2), indicating that the most stable structures actually correspond to the isolated natural arsenicins. (…………..) This could be an explanation, unless the relative abundance of the majority arsenicin A compared to the other arsenicin B-D is due to the presence of the S-reactant as the limiting agent.”
Line 242 kcal, done
- Comment at line 259: I disagree here, see comments above.
this is speculation unless an enantioenriched arsenicin A is being found in nature. At present, racemization is just as fine as a hypothesis as racemic formation, at least on a formal basis.
Answer from the authors
Actually, instead of "This suggests that arsenicin A was isolated in racemic form”, the correct form would be “This suggests that arsenicin A may have been isolated in racemic form”. Thanks to the reviewer's clarification, we have changed the text as:
“Based on this evidence, we can suppose that the use of alcohols and water with possible acid traces in the workup of the sponge extract could have racemized natural arsenicin A, if this was present in an enantio-enriched form. However, it is currently not known whether arsenicin A is produced in pure racemic or enatiomeric form.
Line 272: the low polarity and poor solubility
Line 358: Compound 7 proved
- Comment at line 475: Care should be taken here in the use of models which most certainly have not been applied to compounds of this type before. Lipinski’s rules completely different structures. might be quite misleading as they have been established with
Answer from the authors
We thank the reviewer for this comment, which led us to evaluate the prediction of the main parameters using different software. In particular, we had read that for the calculation of the TPSA SwissADME considers sulfur and phosphorus as polar atoms, but now we have compared PSA and LogP, and BBB score derived by additional Molsoft and Molinsipration tools. Molinspiration methodology for logP calculation is indicated to be very robust and able to process practically all organic and most organometallic molecules. As a reference we also considered the arsenical drug salvarsan.
The values are reported in the additional Table S5, cited inserting the following sentence: " However, due to the peculiar structure of these polyarsenicals, care could be taken in the use these predictive models, which have been developed considering more classic organic molecules. We compared the SwissADME data with those obtained using Molsoft L.L.C. [32] and Molinspiration [33], the latter one indicated as able to process most organometallic molecules. Although expressed in different forms using the three tools (Table S5), the trend for lipophilicity is similar, observing an obvious increase moving from arsenicin A to dimethyl, diethyl and dipropyl analogues 7-9, respectively; the PSA parameters do not distinguish alkylated polyarsenicals from each other, and the BBB permeation result favorable for compounds with ethyl and propyl chains.

Reviewer 2 Report
Based on the unusual marine metabolites arsenicin A-C the authors have made a number of analogues for the targeting of glioblastomal cells. This primarily consisted of adding alkyl chains but also the formation of various sulfur analogues. The compounds gave good activities and presented good ADME properties indicating their potential viability as medicinal agents.
Overall, I found this paper interesting due to the novel nature of the metabolites on which these synthesized structure were based. Given the lack of good treatments for this type of cancer there is an urgent need for better and more effective drugs with reduced side effects.
The study was quite extensive covering many different aspects of drug evaluation – from isolation, synthesis, testing, modeling and evaluation of physiochemical properties. I think it would have been nice to see a few more alkyl analogues made – for example have the authors considered a significantly longer alkyl chain (for example 10 carbons?). Additionally, it might be interesting to use carboxylic acids that contain fluorinated side chains as well. Given this is only one-step synthesis (albeit the yields are quite low); it would be interesting to have slightly more diverse analogues. Maybe these could form the basis of further studies?
It was quite unclear (to me at least) the mechanism of how the alkyl compounds were formed. I looked at the original paper which was in German and did not explain the mechanism. I think it might be good for the reader if the authors explained how these compounds are formed. They could provide the mechanism or at least some indication in the text as to what is going on. It appears that the acids lose one carbon atom but how is not clear and then the oxygen insert (again this is not clear how). This is definitely not a trivial reaction from a mechanistic point of view and a bit of explanation would be greatly appreciated.
Additionally, it appears that compound 7 is missing an “As” atom in the drawing in both scheme 1 and 2. I assume this is the case and not a different mechanism!!! (which refers to my previous point of how these compounds are formed).
Some other minor errors – line 237 has some strange grammar that needs correcting and line 254 I suppose should be “trace acid-catalyzed process” not “acid trace-catalyzed process”
Apart from the small additions mentioned above I would be happy recommend publication of this article.
Author Response
From the authors:
We are very grateful to the reviewer for the time devoted to this revision, as well as for the valuable feedback and constructive comments on the manuscript.The changes are currently highlighted within the new version of the manuscript. Please find below, point-by-point our responses to the specific comments and suggestions.
Based on the unusual marine metabolites arsenicin A-C the authors have made a number of analogues for the targeting of glioblastomal cells. This primarily consisted of adding alkyl chains but also the formation of various sulfur analogues. The compounds gave good activities and presented good ADME properties indicating their potential viability as medicinal agents.
Overall, I found this paper interesting due to the novel nature of the metabolites on which these synthesized structure were based. Given the lack of good treatments for this type of cancer there is an urgent need for better and more effective drugs with reduced side effects.
- The study was quite extensive covering many different aspects of drug evaluation – from isolation, synthesis, testing, modeling and evaluation of physiochemical properties. I think it would have been nice to see a few more alkyl analogues made – for example have the authors considered a significantly longer alkyl chain (for example 10 carbons?). Additionally, it might be interesting to use carboxylic acids that contain fluorinated side chains as well. Given this is only one-step synthesis (albeit the yields are quite low); it would be interesting to have slightly more diverse analogues. Maybe these could form the basis of further studies?
Answer from the authors:
As indicated in the text, the reactions is effectively selective with homologous combinations of linear carboxylic acid and corresponding anhydrides, whereas the steric hindrance of branched reactants prevents the adamantane product formation. We have considered the possibility of obtaining products with longer chains (e.g. 10 carbons), but in these cases the products would have been too lipophilic for biological evaluation. In that case we would have had to resort to carriers, while the dipropyl product proved to be a good structure as it is. Anyway, the length of the chain remains to be modulated as a further perspective. Based on the relevance of the fluorine atom present in molecules of interest for medicinal chemistry, we thank the reviewer for its suggestion to use carboxylic acids containing fluorinated side chains.
- It was quite unclear (to me at least) the mechanism of how the alkyl compounds were formed. I looked at the original paper which was in German and did not explain the mechanism. I think it might be good for the reader if the authors explained how these compounds are formed. They could provide the mechanism or at least some indication in the text as to what is going on. It appears that the acids lose one carbon atom but how is not clear and then the oxygen insert (again this is not clear how). This is definitely not a trivial reaction from a mechanistic point of view and a bit of explanation would be greatly appreciated.
Answer from the authors:
As pointed out by the reviewer, it is not a trivial reaction from a mechanistic point of view and Keppler did not even give any details or hypotheses on the mechanism. Regarding the synthesis of arsenicin A, obtained in mixture with its adamantane analogues, we reported (Mancini et al. Sci Rep 2017): "To look deeper into the intriguing mechanism of this reaction, we verified that the reaction did not occur without K2CO3 or even in the absence of acetic anhydride. Whereas, the reaction proceeded quite well without the acetic acid, or when CH3COOH/K2CO3 was replaced with sodium acetate. This is reminiscent of the mechanism involved in the formation of cacodyl (=tetramethyldiarsine) and cacodyl oxide, which are components of Cadet's fuming liquid (Seyferth, D. Organometallics 2001, 20, 1488).”
The mechanism could involve a nucleophilic attack on the electrophilic arsenic with decarboxylation, possibly acting on the arsenolite As4O6, a crystalline adamantane structure deriving from arsenic trioxide. However, at the moment we have no further evidence and therefore we prefer not to report any hypotheses on the mechanism. We reserve to go into more insights in a future work, possibly developing the study on longer and/or modified chain analogues.
- Additionally, it appears that compound 7 is missing an “As” atom in the drawing in both scheme 1 and 2. I assume this is the case and not a different mechanism!!! (which refers to my previous point of how these compounds are formed).
Answer from the authors:
One As atom was missing
- Some other minor errors – line 237 has some strange grammar that needs correcting and line 254 I suppose should be “trace acid-catalyzed process” not “acid trace-catalyzed process”
Answer from the author:
Line 237 : …. could lead to the assumption of a higher stability
Line 254: done
Apart from the small additions mentioned above I would be happy recommend publication of this article.

Reviewer 3 Report
Mancini and coworkers describe a chemical and biological investigation of a group of poly arsenicals active against glioblastoma (GBM) cell models. The work is impressive by amount of experimentation and attention to detail. The results are also very promising as these compounds show interesting activity against of a panel of GBM stems and high selectivity in their poor effects against non-cancerous cells. This reviewer recommend acceptance with just a minor revision. Compound 7 appears to have been synthesized and evaluated before, yet no comparison with the previous results is drawn. Perhaps in Conclusion section some mention could be made on whether the new results are consistent with the previous ones and in what way.
Author Response
From the authors:
We are very grateful to the reviewer for the time devoted to this revision, as well as for the valuable feedback and constructive comments on the manuscript.
The changes are currently highlighted within the new version of the manuscript. Please find below, point-by-point our responses to the specific comments and suggestions.
Mancini and coworkers describe a chemical and biological investigation of a group of poly arsenicals active against glioblastoma (GBM) cell models. The work is impressive by amount of experimentation and attention to detail. The results are also very promising as these compounds show interesting activity against of a panel of GBM stems and high selectivity in their poor effects against non-cancerous cells. This reviewer recommend acceptance with just a minor revision.
- Compound 7 appears to have been synthesized and evaluated before, yet no comparison with the previous results is drawn. Perhaps in Conclusion section some mention could be made on whether the new results are consistent with the previous ones and in what way.
Answer from the authors:
No comparison with the previous results was made for compound 7 because it had been evaluated on human tumor cells which did not include those of glioblastoma or glioma in general, in any case no results on tumor stem cells.
At line 75 of the text it is reported. “Also the tetraarsenical adamantane structure 5-7 (Figure 2) obtained by Mancini et al. and selected for an in vitro screening on the NCI- full panel of human cancer cell lines showed significantly higher cytotoxicity compared to ATO against all the various cancer cell lines tested [12], with compound 7 being particularly effective in inhibiting growth of solid tumor cell lines”
In conclusion section, the biological activities of 7 have been mentioned as follows: “Compound 7 had previously been evaluated as an efficient inhibitor of the growth of a large series of tumor cells, but not including glioblastoma, nor tumor stem cells.”
From the authors:
We are very grateful to the reviewer for the time devoted to this revision, as well as for the valuable feedback and constructive comments on the manuscript.
The changes are currently highlighted within the new version of the manuscript. Please find below, point-by-point our responses to the specific comments and suggestions.
Mancini and coworkers describe a chemical and biological investigation of a group of poly arsenicals active against glioblastoma (GBM) cell models. The work is impressive by amount of experimentation and attention to detail. The results are also very promising as these compounds show interesting activity against of a panel of GBM stems and high selectivity in their poor effects against non-cancerous cells. This reviewer recommend acceptance with just a minor revision.
- Compound 7 appears to have been synthesized and evaluated before, yet no comparison with the previous results is drawn. Perhaps in Conclusion section some mention could be made on whether the new results are consistent with the previous ones and in what way.
Answer from the authors:
No comparison with the previous results was made for compound 7 because it had been evaluated on human tumor cells which did not include those of glioblastoma or glioma in general, in any case no results on tumor stem cells.
At line 75 of the text it is reported. “Also the tetraarsenical adamantane structure 5-7 (Figure 2) obtained by Mancini et al. and selected for an in vitro screening on the NCI- full panel of human cancer cell lines showed significantly higher cytotoxicity compared to ATO against all the various cancer cell lines tested [12], with compound 7 being particularly effective in inhibiting growth of solid tumor cell lines”
In conclusion section, the biological activities of 7 have been mentioned as follows: “Compound 7 had previously been evaluated as an efficient inhibitor of the growth of a large series of tumor cells, but not including glioblastoma, nor tumor stem cells.”
From the authors:
We are very grateful to the reviewer for the time devoted to this revision, as well as for the valuable feedback and constructive comments on the manuscript. The changes are currently highlighted within the new version of the manuscript. Please find below, point-by-point our responses to the specific comments and suggestions.
Mancini and coworkers describe a chemical and biological investigation of a group of poly arsenicals active against glioblastoma (GBM) cell models. The work is impressive by amount of experimentation and attention to detail. The results are also very promising as these compounds show interesting activity against of a panel of GBM stems and high selectivity in their poor effects against non-cancerous cells. This reviewer recommend acceptance with just a minor revision.
- Compound 7 appears to have been synthesized and evaluated before, yet no comparison with the previous results is drawn. Perhaps in Conclusion section some mention could be made on whether the new results are consistent with the previous ones and in what way.
Answer from the authors:
No comparison with the previous results was made for compound 7 because it had been evaluated on human tumor cells which did not include those of glioblastoma or glioma in general, in any case no results on tumor stem cells.
At line 75 of the text it is reported. “Also the tetraarsenical adamantane structure 5-7 (Figure 2) obtained by Mancini et al. and selected for an in vitro screening on the NCI- full panel of human cancer cell lines showed significantly higher cytotoxicity compared to ATO against all the various cancer cell lines tested [12], with compound 7 being particularly effective in inhibiting growth of solid tumor cell lines”
In conclusion section, the biological activities of 7 have been mentioned as follows: “Compound 7 had previously been evaluated as an efficient inhibitor of the growth of a large series of tumor cells, but not including glioblastoma, nor tumor stem cells.”
